# Listening to Indigenous Voices, Interests, and Priorities That Would Inform Tribal Co-Management of Natural Resources on a California State University Forest

Zachary J. Erickson [1,*], Kevin Boston [1], Michael J. Dockry [2] and John-Pascal Berrill [1]

1 Department of Forestry, Fire, and Rangeland Management, California Polytechnic State University Humboldt, 1 Harpst St., Arcata, CA 95521, USA
2 Department of Forest Resources, University of Minnesota, St. Paul, MN 55108, USA
* Correspondence: zje13@humboldt.edu

**Abstract:** Indigenous communities have experienced a loss of access and ability to contribute to the management of natural resources due to removal from lands, marginalization, and conflicting knowledge systems. Currently, there is increasing momentum toward re-engaging tribes as stewards of their ancestral lands. This article outlines tribal views on co-management and identifies the forest management objectives of a tribal partner to help better inform a forest co-management partnership between a Native American Tribe (Wiyot Tribe) and a California Polytechnic State University (Humboldt). Qualitative research methods were used to analyze 13 semi-structured interviews utilizing an adaptive co-management framework with enrolled tribal members and representatives to understand the expectations and perceived barriers to a successful co-management relationship. Interviewees repeatedly mentioned interest in the management of wildlife, forest health and resilience, and fuels reduction. Participants also expressed interest in incorporating education and training of tribal youth in the management of forest resources and traditional ecological knowledge. The semi-structured interviews provided participants a platform to share their thoughts and express their feelings regarding the future stewardship of ancestral forest lands.

**Keywords:** traditional ecological knowledge; forestry; collaboration; indigenous land management; university partnerships; adaptive co-management

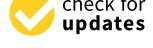



## 1. Introduction

Indigenous communities have lost access to and their ability to contribute to the management of natural resources within their ancestral lands. Two factors contributing to this loss are marginalization, and conflicting knowledge systems [1]. Some believe that the absence of indigenous involvement has led to a decline in both the quality and abundance of culturally important resources [2–4], as well as limited the intergenerational transfer of traditional ecological knowledge, or TEK [5]. Reasons for the decline of cultural resources include the suppression of wildfire and a reduction in cultural burning [6–10]. It follows that by re-engaging tribes as stewards in their ancestral lands we can foster and develop TEK and improve culturally important resources. One such pathway involves developing collaborative management, or co-management, partnerships. In the United States, there is increasing momentum for the formation of these partnerships on public lands [11,12].

### 1.1. Tribal Co-Management of Natural Resources

Though California has the largest population of Native American tribes in the United States [12] it ranks among the lowest in terms of the amount of land under tribal control in comparison to other states [13]. A Statement of Administration Policy [14] recently released by the California Governor's Office uses explicit language supporting the co-management of lands under the ownership of the State of California. This statement is supported by a

string of related Executive Orders that acknowledged past harms done to Native American tribes [15] and created a network of tribal liaisons in California's public land agencies [16] to facilitate consultation and collaboration. Further adding to the momentum of tribal co-management is California's "30 by 30" initiative effort which identifies TEK as an important tool and urges agencies to collaborate with Native American Tribes to "better understand our biodiversity and threats it faces" [15].

At present, there appears to be limited case study examination of university—tribal collaborative partnerships [17,18]. In the U.S.A., public universities are state agencies, but many individual universities have limited lands to manage. This fact, combined with the educational missions of universities, provides a unique opportunity and premise for the formation of tribal co-management partnerships.

The term co-management first arose during the late 1970s from USA treaty tribes in the Pacific Northwest describing their desire for shared decision-making authority and regulation of fisheries in the Columbia River Basin [19]. This resulted in the formation of the Columbia River Inter-Tribal Fish Commission in 1977 [20]. Co-management has since been used to describe a variety of other collaborative arrangements and lacks a single definition [21]. Researchers broadly define co-management as the sharing of decision-making authority and responsibility between government agencies and local resource users [22]. Early cases of co-management were largely formed around the decentralization of fisheries management [22,23]. Co-management has since expanded to other common-pool resource management such as forestry [24,25] and wildlife [26]. This management arrangement has also been presented as an adaptive approach to addressing complex social-ecological issues such as the impacts of climate change [27,28].

To reach "complete co-management", agencies and universities will need to reach a place of shared decision-making authority in the management of public lands with Native American tribal partners [19,29,30] but in practice, this has rarely occurred [31,32]. Since the emergence of the term, co-management has been described as having varying levels of participation [1,22,29,33,34] that can be likened to climbing rungs on a ladder [35]. But complete co-management disregards the lower rungs of participation from tribal and citizen partners such as consultative [29,35], instructive [34], and informing [22,33,35] relationships. These lower levels of participation have been described as tokenism [35]. The upper rungs of this ladder are identified as partnerships [29,35] and joint action [29] which more closely resemble the type of relationship to which the term co-management was first applied.

According to researchers, co-management should be viewed as an exercise in building long-term relationships [21,36,37] to improve management rather than a technique to simply manage natural resources [38,39]. To this end, co-management has been viewed as building and maintaining trust [40,41], building social capital [39,42,43], and not as an end goal of the partnership [32,44].

Cases of co-management have been documented in the United States [18,24,45] and around the world [46–48]. Components of successful co-management cases include developing informal relationships [24,49], building trust [50], accounting for colonial legacies [17,18,24], identifying mutual benefits [51], and mutual learning [18,26,52]. There are a few components of "tribal" co-management which make it distinct from research partnerships built with non-native partners. Sowerwine et al. [17] highlight a few of these priorities which include a focus on whole ecosystem management and supporting youth empowerment. Researchers in education have noted that working effectively with tribal youth requires an understanding and consideration of their culture [53–57].

### 1.2. Adaptive Co-Management of Natural Resources

Adaptive co-management (ACM) includes elements of both adaptive management [58] and co-management [37,39]. The definition of ACM broadens the scope of the co-management definition offered by Berkes [22] to emphasize the importance of including an iterative process of reflection and the process of learning by doing [59]. Researchers highlight that ACM should not be perceived as an end goal, but rather a process of "negotiation,

deliberation, knowledge generation and joint learning" [39]. Co-management relationships where social learning does not occur are likely to fail [39]. While researchers recognize that ACM is not a panacea [37,60,61], it has proven to be a useful framework for tackling complex issues involving multiple stakeholders [27,28,62,63].

Enabling environments, including both institutional and legislative, are considered bridges [64] and cornerstones to developing ACM partnerships [39,65,66]. Early stages in ACM are defined by the transition from stakeholders acting independently to entering dialogue and initiating learning processes [28,37]. Working to ensure positive experiences at this formative stage of the partnership can yield financial and social dividends in the future [67].

ACM relationships are based on the principles of social and institutional learning [68–70] that, within ACM, can encourage partners from different knowledge systems to co-generate new knowledge regarding natural resources management [39,71]. While the integration of TEK into ACM and thus western ecological knowledge (WEK) has been receiving interest as a potential tool to face complex social-ecological management issues [26,52,72], several researchers have noted a shortfall of fruitful outcomes when integrating indigenous perspectives into management decisions [31,73–75] and coopting local knowledge from agency partners [40]. There are also issues tied to conflicting missions or values [49,50] and a bias toward WEK-derived plans over those that incorporate TEK [24,31,49].

### 1.3. Informing Co-Management of a University Forest

The Wiyot Tribe is a federally recognized tribe in northwest California whose ancestral lands stretch from Bear River Ridge to the south and Little River to the north (Figure 1). Before European colonization, it is estimated that the region supported between 1500 and 2000 inhabitants [76]. After a long history of genocide, relocation, and dispossession of their ancestral lands, the Wiyot Tribe was recognized by the federal government in 1908. In 1961, the Wiyot Tribe was terminated under the California Rancheria Act and did not regain federal recognition until 1981 through a federal lawsuit. By this time, they had limited access to their ancestral lands where the title was held predominately by private owners but included state and local government agencies [76] (Figure 1).

The forest provides cultural resources important to the Wiyot Tribe. The coast redwood (*Sequoia sempervirens*) is an integral resource for the construction of both canoes and houses [77]. Understory plants such as the red huckleberry (*Vaccinium parvifolium*), evergreen huckleberry (*Vaccinium ovatum*) and beaked hazelnut (*Corylus cornuta*) are important foods gathered by the Wiyot Tribe. Beaked hazelnut stems are also used in the weaving of baskets as well as fishing equipment [78].

In 2018, the California Polytechnic State University Humboldt (CPH), located in Arcata, north coastal California, U.S.A. was deeded from R. H. Emmerson & Sons, an approximately 360-ha redwood-dominated (*Sequoia sempervirens*) forest located 12 km southeast of CPH campus at the headwaters of Jacoby Creek, or Goukdi'n which is the indigenous Wiyot Tribe's name for that area of their ancestral lands (Figure 1).

Before university ownership, the Jacoby Creek (Goukdi'n) Forest was actively managed for timber production by Sierra Pacific Industries. The forest comprises a mix of conifer and hardwood tree species and age classes. The forest straddles Jacoby Creek, a fish-bearing stream, and contains remnant old-growth western red cedar (*Thuja plicata*). The property title is currently held by a public institution, the California State University Board of Trustees, and has stated goals of preservation and protection of the forest, improved wildlife habitat, and water quality, and providing opportunities for research and education [79].

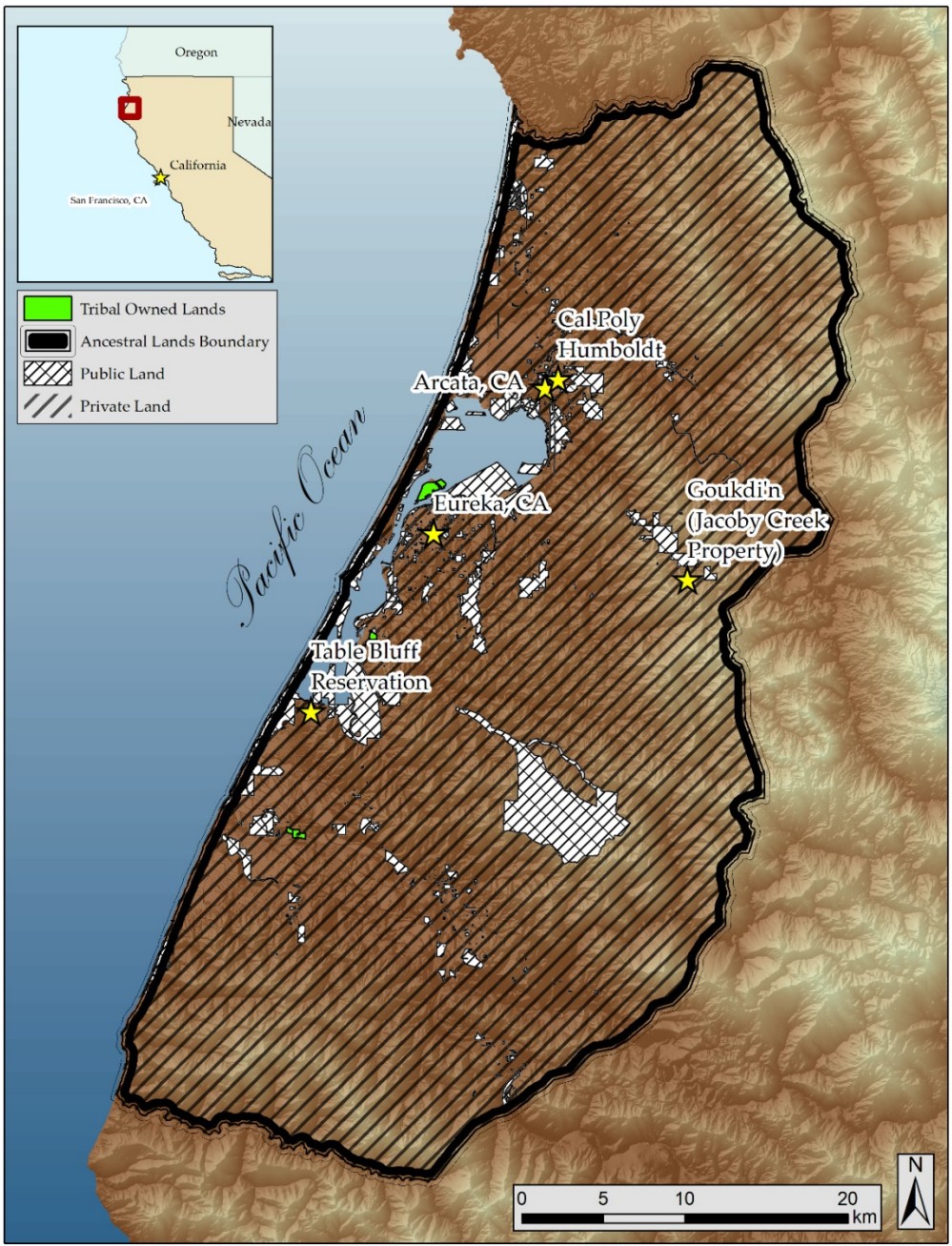

**Figure 1.** Wiyot Tribe ancestral lands in the northwest corner of California, USA (black border). Private lands (black stripes) within Wiyot Tribe's ancestral lands encompassing the vast majority of the area. Stars indicate two cities, Arcata and Eureka, as well as the Cal Poly Humboldt campus, Goukdi'n, and the Table Bluff Reservation. Green polygons indicate land owned by the Wiyot Tribe. Grey polygons with crosshatching indicate public lands within the Wiyot Tribe ancestral lands boundary. The inset map locates the Wiyot Tribe ancestral lands (red box) in the northwest corner of California.

The Wiyot Tribe and CPH have collaborated through the development of the Sea Level Rise Institute (SLRI). The SLRI is an ongoing collaboration between university, tribal, non-governmental organizations, community, and agency partners to "reimagine the future of the California coast" in the face of rising ocean levels [80]. In 2020, CPH (known at the time as, Humboldt State University), developed a 5-year strategic plan which placed tribal collaboration and relationship building as one of its highest priorities [81]. With this decree

to work with local Native American tribes combined with the recent acquisition of the Goukdi'n property, there was an opportunity for both CPH and the Wiyot Tribe to develop a co-management partnership. This partnership would provide benefits to the Wiyot Tribe and CPH including place-based education, training, and research into traditional and western ecological knowledge and practices, and forest management activities that incorporate tribal values.

The goal of our study was to elucidate the views of a tribal partner in the early stages of building a co-management relationship. To do this, we conducted semi-structured interviews to determine the goals that various tribal members would have for forest management. This method of data collection is used to highlight the importance of the individual without diminishing the prospect of community and collaboration [82]. This method of data collection has been utilized either in conjunction with other qualitative methods [74] or by itself [49,52,83,84]. We chose this data collection method over other methods such as surveys or focus groups because we wanted to understand the views of individual tribal members as well as initiate dialogue between the university and the tribe. Our three research objectives were to understand members of the Wiyot Tribe's (i) views on co-management in general, and specifically (ii) their forest management goals in Goukdi'n, and (iii) the importance of engaging tribal youth in natural resource management. This investigation contributes information that can be used by CPH and also by similar institutions that are in the early stages of forming relationships with tribal partners for the co-management of natural resources and to support the future development of co-management theory by understanding why parties participate in collaborative processes [32].

## 2. Materials and Methods

This study uses applied-qualitative analysis which is research that contributes to the understanding of a contemporary issue [85] under the iterative and adaptive framework of Grounded Theory Analysis [86]. We utilized semi-structured interviews [46,49,84,87,88] based on an interview guide (Supplementary Materials Table S1) that contained open-ended questions seeking to understand participant's views on co-management, forest management goals in Goukdi'n, and the importance of engaging tribal youth in natural resource management.

Presenting the project proposal to Wiyot Tribal Council was the first step in this project. After obtaining approval from the council, the proposal was then submitted to the university's Institutional Review Board (IRB 20-124). To initiate dialogue between the university and the tribe about co-management, we attended meetings open to the public held by the Wiyot Tribe Natural Resources Department (NRD) where we were able to present methods and goals for the research and also recruit the first participants for the study. These interactions included two site visits to Goukdi'n to build the relationship and discuss the potential co-management partnership.

Following decolonized research methodologies (DRM) [89] and similar co-management case studies [17,18], this project sought Wiyot Tribal Council approval and worked to inform the tribe throughout the research process [18,84]. Periodic updates were provided to the tribe through tribal council meetings including alerting them to presentations and sharing current results. Before publication, this manuscript was presented and submitted to the tribal council for review. Research methods including participant confidentiality, participant recruitment, and sampling procedures were approved by the IRB. These procedures include the protection of participant identity through the coding of direct identifiers [52] and codebooks stored in password-protected locations.

Purposive sampling in conjunction with the snowball method [90,91] identified potential participants that could provide an opportunity for in-depth exploration of the project research questions [85,92]. These participants were first identified through key informants with contacts within the Wiyot Tribe and by attending tribal meetings open to the public. With no previous relationships or contacts within the tribe, we anticipated that this would

increase the variation in participants by reaching different families and social circles. Difficulties in participant recruitment were exacerbated by the various COVID-19 lockdowns from 2020–2022.

Interviews were conducted in person, over the phone, and through Zoom Inc. remote meeting software. Each interview was recorded and transcribed through the Otter.ai transcription software. Transcripts were verified and imported to Atlast.ti 9 qualitative data analysis software within 48 h of interview completion. Transcripts were read a minimum of three times utilizing line-by-line, axial, and focused coding [86] which used higher-order categories defined by the three research questions [52]. Code occurrences were analyzed in two ways. First, we identified the number of participants that shared information on a theme or topic. Second, we normalized the absolute and relative frequency of responses to adjust for differences in transcript length. Once normalized, we then examined the code frequency occurrence based on our themes to identify codes and themes that emerged with greater frequency than others. This allowed for an establishment of a hierarchy of goals and expectations from participants. The participant sample size was considered adequate when no new themes or information relating to the research questions emerged [93].

## 3. Results

We interviewed 13 enrolled members and representatives of the Wiyot Tribe. Interviews took place from July 2021 to August 2022. The interview length ranged from approximately 30 min to over 120 min. Participant age ranged from 20 to 71 years old with an average age of 40. Mostly women were interviewed ($n = 10$) and all but one participant were members of the Wiyot Tribe. The single non-tribal member held extensive knowledge of natural resource management and was employed by the Wiyot Tribe NRD. Of the participants interviewed, 37% had experience working in some form of natural resource management (Table 1), while none had specific professional forestry experience.

**Table 1.** Demographic breakdown of interview participants.

| Average Age (Years) | Gender | Experience in Natural Resources? | Tribal Member? |
| --- | --- | --- | --- |
| 40 (Min 20, Max 71) | Female (70%) Male (30%) | Yes (37%) No (63%) | Yes (92%) No (8%) |

Although participants were asked only general research questions, important themes emerged that highlighted a need for understanding the history of the Wiyot Tribe and their relationship to their ancestral lands. The following subsections provide summaries of data collected through semi-structured interviews by theme and are accompanied by direct quotes to support each theme [49].

### 3.1. Regaining Access to Ancestral Forests Can Promote Cultural Knowledge and Reconnect Tribal Members as Stewards of the Land

*"We were severed from the forest. When people realized there was gold in the hills and that . . . the redwood trees were just as lucrative as gold . . . we were severed from that relationship without our permission, you know, without our input. And I think that being able to finally have a forest again could reclaim and revitalize that traditional cultural knowledge, the traditional ecological knowledge."*

#### 3.1.1. Tribal Capacity

Tribal capacity was a topic brought up by 92% of participants ($n = 12$). Participants shared that they viewed this co-management partnership as an opportunity to develop programs that could build tribal capacity in forestry and fisheries management. Citing their forced disconnection from the land for over 100 years, participants also shared that they don't currently have expertise in forestry themselves. When discussing how being

disconnected from the land has impacted tribal capacity one participant shared, "*I don't know a lot of the cultural perspectives on that because . . . those things were taken away.*"

They shared that although they appreciate their non-Wiyot NRD employees, there is currently a lack of Wiyot Tribe citizens working in the department and they currently don't have trained foresters on staff. Participants suggested that the university could offer paid positions, stipends, or an endowment to ensure that the tribe could be involved in the co-management of Goukdi'n. When discussing these potential opportunities, one participant presented the idea of "*building up an endowed Wiyot position or two positions that is, basically, shadowing and the assistant to the forest manager.*"

Tribal capacity was also identified as a potential barrier to co-management. Limitations in staff, knowledge and technological resources were each indicated to be something that would need to be addressed in the co-management partnership. Capacity is already strained by responsibilities such as reviewing Timber Harvest Plans (THP) and employees already fulfilling multiple roles. Participants contrasted the small Wiyot Tribe NRD against large companies or the university that have larger budgets. "*. . . in the cultural department, it's Ted [Tribal Chairman], who has an assistant who . . . assists him sort of reviewing projects, timber harvest plans but also is dually placed over in our childcare center or childcare program.*"

One participant shared that this can sometimes make it seem like the tribe doesn't care, but this perceived lack of interest due to limited capacity, "*it's not our unwillingness to, to steward these lands. But it's, it's our ability to just not have resources and people power. Honestly, that's it.*"

### 3.1.2. Accessing Ancestral Lands

Accessing ancestral lands was shared as a major benefit of this co-management partnership by 85% of the participants (*n* = 11; Figure 2). When discussing access, participants shared that there are limited forested lands that the tribe has access to, and they currently do not feel welcome to practice cultural activities on private lands within their ancestral boundaries.

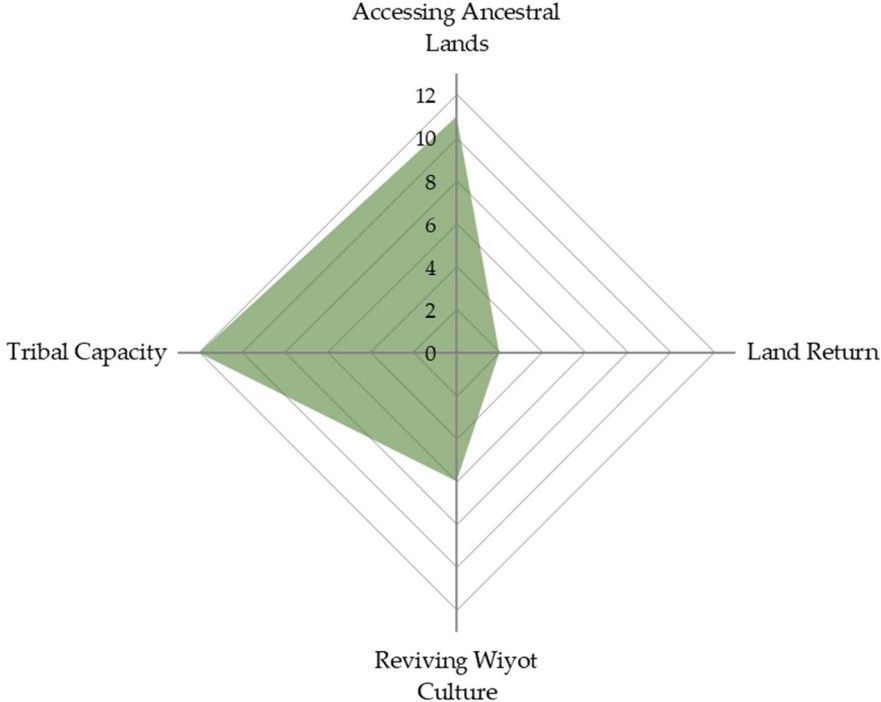

**Figure 2.** Subthemes that are within the broader theme of "Regaining Access to Ancestral Forests Can Promote Cultural Knowledge and Reconnect Tribal Members as Stewards of the Land". Subthemes of 'tribal capacity' and 'accessing ancestral lands' were mentioned by the most participants with 12 and 11, respectively.

Participants shared fears of "*hopping fences*" and "*trespassing*" on private lands. One participant shared that on public lands where they do have access, they have to contend or compete with all other members of the public using these resources. "*I can't even hunt in my own lands, so I have to go and try to fight it out with the rest of the people in the forest.*"

When discussing activities that could be supported by access, participants particularly focused on being able to gather medicines, traditional foods, and basketry materials. Participants also shared other benefits to access including activities such as camping or just having a place where a tribal member could connect back to the land. This included having ceremonies, a village site, and a space where they would feel welcomed to "*go be themselves.*"

Regaining access was shared as being a positive for passing on knowledge, especially to younger generations. Participants indicated that they have been "severed" from their ancestral lands and that because of this change, "*...a lot of our traditions have, you know, maybe not survived this whole time.*" One participant was afraid of this detachment of identity and was interested in building a place that was inviting to the tribe that would "*...allow Wiyots to walk in peace.*"

Many shared statements did not focus on a single activity and instead said that co-management would offer a space for the tribe to get out into the forest and get together. Of the participants, 23% (*n* = 3) compared their lack of access to that of surrounding tribes such as the Yurok and Hoopa. Participants shared that most of the Wiyot Tribe's lands are held privately (Figure 1) while others are surrounded by federal land. One participant shared that tribal members need to travel over an hour to get to public lands where they could hunt.

### 3.2. Conditions for Achieving True Co-Management

By interpreting and assigning quotes taken from participant transcripts into sub-themes, we found that participants had insightful opinions on particular opportunities and challenges of implementing co-management. Important co-management subthemes related to forest access, roles and responsibilities, decision-making authority, preference for collaboration as opposed to consultation, and benefits of management (Table 2).

**Table 2.** Examples of the assignment into subthemes of quotes relating to the theme of co-management are taken from participant transcripts.

| Subtheme | Quote |
| --- | --- |
| Provide Access | "*You know we—all forests are either held privately or through the state or feds and although we can, you know, access these places it's with permissions, right? It's always with somebody else's permission and if we co-manage this land, I don't want to have to ask permission.*" |
| Adaptive Roles and Responsibilities | "*...I think it will be different. Each project will be different. Each one comes with a different mindset.*" |
| Sharing Decision-Making Authority | "*Well, the university has to get off the high horse, because they are the ones because it's their university, it's their property. And they are going to have to concede some of their authority because it's theirs. And we've got to remember that it's theirs. It's not ours. So they have to concede some of their authority so that we can meet them at least halfway and talk.*" |
| Moving Beyond Consultation | "*So, that's one of the components, is that . . . it's an ongoing—I don't want to say consultation–it's an ongoing decision-making group that we're part of. And that we aren't just there to consult. That we're there to manage just as well as the other groups of people or the other representatives.*" |
| Identify Mutual Benefits in Management | "*But...with a collaboration, you know everybody's committing together. We're giving our word that we're going to arrive at this location we're going to talk about the management of this particular forest, we're going to develop a forest management plan that supports its health and well-being both for the forest, as well as Wiyot people, as well as you know forestry students.*" |

### 3.2.1. Listening, Learning, and Respecting Tribal Perspectives

*"Well, I think that it means, again, as a group, I guess understanding that the knowledge that indigenous people hold of lands and waters and critters is based on thousands upon thousands of years, of living in this location and that it should be held as primary evidence of . . . stewardship and land management."*

Most participants (92%) acknowledged that there would be a difference in opinions, but most were optimistic that university partners would be open to tribal perspectives and priorities (Figure 3; subtheme: 'having an open mind, listening, and learning together'). Another shared that, as Native Americans, they don't always think like the university. *"We don't always think like the university or like the government, or like private citizens who own private land that was once our land."*

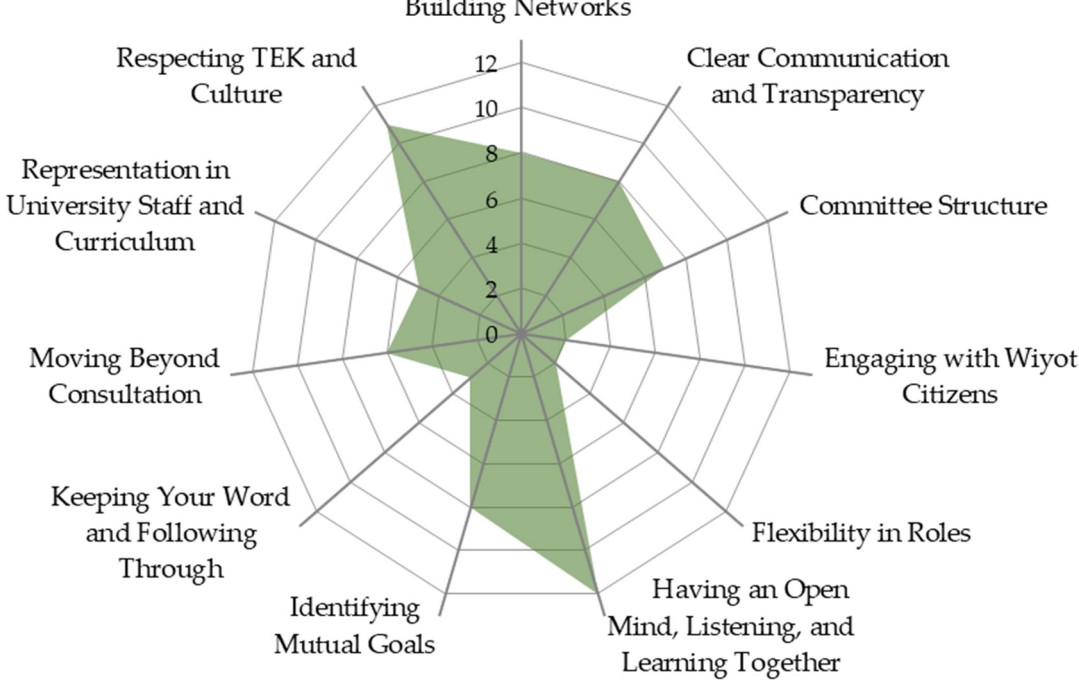

**Figure 3.** The frequency of participants that spoke about subthemes relating to co-management. For example, 8 out of 13 total participants spoke about the co-management subtheme 'building networks'.

One participant indicated that they have little faith that the university can respect Wiyot Tribe TEK and that it will be, "thrown out the window because of their ideology...". Others were more optimistic about the partnership indicating that the university and tribe should, *"Just listen to what each of the groups have to say. And then we should be able to work together for the project . . . ."*

They also shared that there will be learning and the exchanging of ideas on both sides, not just the tribe teaching the university, or vice-versa. Participants shared that TEK, such as cultural burning, has been ignored for years and the university should be open to learning about their ways of managing it, *" . . . we learn something new, we share the knowledge, and we bring people in, and we share the knowledge and all the while working together."*

Participants also indicated that they understand that there will be a compromise, or "give and pull", on both sides while managing the property. This requires some sharing of power, which one participant shared might be difficult for some university partners, *" . . . when you come to the table, you come to the table and you know, you're going to have to give up something. Both sides. There's a give and pull."* One participant shared that disagreements should not lead to university partners giving up on the partnership and not saying to the tribe *"This is our project...you stay out of it."* They hoped that the university would understand the tribe's connection to the land and try to *"...support that to the best of their*

*knowledge*." There is an expectation that this knowledge is held primary and that university partners are willing to incorporate it into WEK.

When discussing the issue of power, one participant indicated that letting go of it can be scary for some people, but they hope they can support them in learning new information. Another participant shared that it is important that the university not overrule the tribe because that could lead to more disagreement. *"It could be really top heavy from the Cal Poly side, or the Wiyot Tribe side so, like really trying to understand that it's a collaboration and partnership, not one trying to supervise the other."*

### 3.2.2. Moving beyond Consultation toward Shared Decision-Making

*"So, I mean a collaborative relationship is much more involved. It's, it's ongoing, it's–it continues it doesn't stop. It considers all voices at the table. Whereas with consultation, you know again it's just a letter to, to [tribal member] and [tribal member] writes off on it and, and it may or may not happen how [tribal member] wants it to happen."*

Participants shared that while the university holds title to the land, they would have to give up some decision-making authority in the co-management partnership. This includes not just asking the tribe to sign off on things or just giving presentations but rather being a managing partner at an *"equal level"* with the university. This sentiment was shared by one participant who shared that the university would need to " . . . *concede some of their authority so that we can meet them at least halfway and talk*."

The partnership was viewed by many participants as being a long-term commitment. When describing the length of the relationship they used terms such as *"ongoing," "forever,"* or *"no sunset."* Another participant shared that within this long-term relationship, decision-making authority may not always be the same. A participant shared that authority would be a compromise and they would need to find an *"equal spot, that we're comfortable with at both sides."* When asked if this would always be at the same spot, they responded, *"No, I think it will be different. Each project will be different."*

### 3.3. Centering Forest Management Activities on Cultural Resources, Wildlife Habitat, and the Restoration of Natural Processes

Specific research projects that could occur in the forest were suggested by 38% (*n* = 5) of the participants (Figure 4). Some projects were forestry specific, including the treatment of underbrush, vegetation mapping, and reforestation techniques. Other potential projects related to water quality and fisheries. When discussing a potential research project, one participant with experience in natural resources shared how treatments could be tested to assess their ability to promote native plant species by, *"cutting down all of the underbrush around these trees, just like everything once a year, then seeing how it grows back better and also doing like burning having a burn, seeing if that helps things for like getting rid of the ivy or getting rid of any non-native plant any non-native plants . . . "*

The terms *"sustainable," "ethically,"* and the phrase " . . . *it could be done better*," were used when discussing how harvesting should be done. Six participants (46%) viewed clear-cutting, specifically, as negatively impacting wildlife habitat and was visually displeasing. Two of the participants in this study (15%) indicated that they would not want to see any harvesting in the forest, but the majority saw timber harvesting as a *"necessary evil."* When asked their thoughts on timber harvesting, one participant shared, *"It can be done, and has been done, ethically and responsibly. It's a necessary evil. Like, replanting where they're harvesting is nice. No, I mean, as long as it's being done ethically and responsibly, I don't have a problem with it."* One participant indicated that they do not think any harvesting is necessary for the forest. All other participants viewed some form of active management as necessary to improve forest health, improve wildlife habitat, reduce excessive stand densities, and reduce wildfire risk (Figure 4).

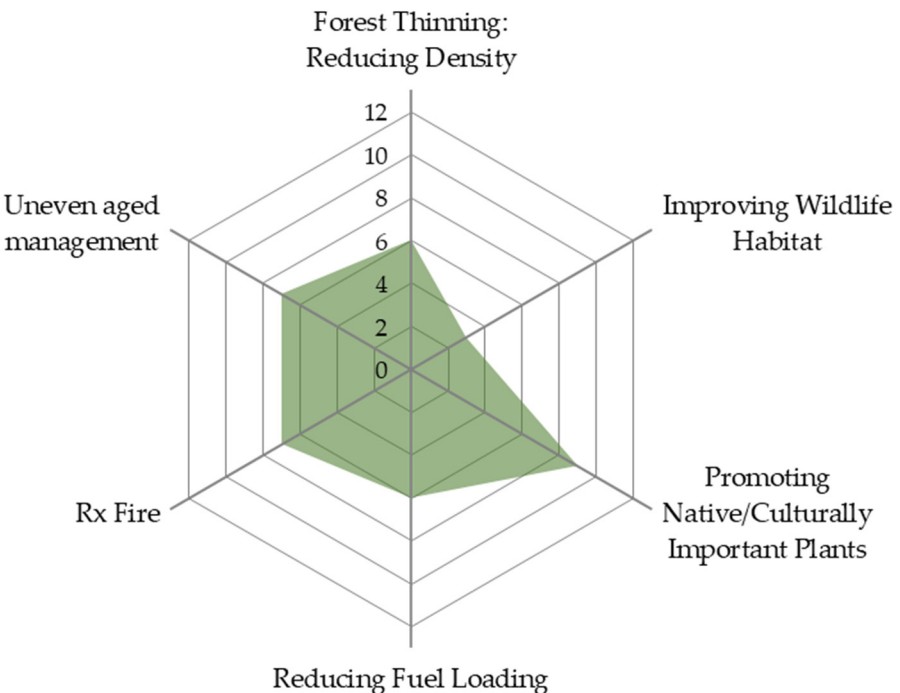

**Figure 4.** Within the subtheme of "Active Management", this graph indicates the number of participants who spoke about each management activity. For example, the subtheme 'promoting native/culturally important plants' was talked about by 9 participants.

Participants shared that the forest may need to be thinned to remove dead and dying trees but also highlighted the importance of dead trees for wildlife habitat (Figure 4). There was also a focus on removing diseased trees with a particular focus on sudden oak death due to its impact on tanoak (*Notholithocarpus densiflorus*), a culturally important food to Native Americans on the western coast of the United States. When discussing tree harvesting, a participant shared thoughts on thinning diseased trees, "*This is a diseased tree, we need to cut it out, burn it, get rid of it so the other trees don't get infected. I just know things like that. That's just common sense. Because you can't make board feet out of a diseased tree. Right?*"

In combination with thinning, participants were interested in utilizing controlled burns, or cultural burning, as a way to "clean up" the forest, reduce invasive species, and promote wildlife habitat. One participant shared that Native Americans had been concerned about the risk of wildfire for years, "*...Native Americans have been predicting it for years...when this starts to burn...this is gonna burn everything up. I've been hearing that since I was a kid. You know, like they've seen it coming...because they saw the mismanagement.*" Another participant shared how fire was required for certain basketry materials, "*...look at beargrass. Beargrass needs to be burned before . . . you can gather and weave it,*" and to maintain prairies, "*But then after that burns, guess what? The elk love to come in there, the deer come in there and they roam through their meadows. Bobcats out there, coyotes, fox.*" These participants were making connections between fire exclusion, fire use, and the promotion of culturally important species like basket material and wildlife (Figure 4).

Regarding the potential for uneven-aged management mentioned by 54% of participants, one shared that they would like to see some trees felled to provide light resources for understory trees while another explicitly wanted to see "*different size-class trees,*" and the preservation of western redcedars and hemlocks. They are also interested in seeing trees planted either before harvest occurs or after the removal of trees. A participant with experience in natural resources shared their idea of a well-managed forest as "*...beautiful and open, lots of diversity, lots of different size-class trees, but lots of mature and maturing trees, distinct plant communities, really rich riparian corridor and going upslope, you know, trying to preserve the western redcedars and hemlocks.*" The idea of not harvesting all trees was expressed in terms

of "*only taking what you need*" when removing resources. Another participant elaborated, "*Yeah, do it in sections. Like just don't do it all in one spot. I was taught that from a young age like when you go and gather you don't take the whole plant. So that way, it could thrive and come back to life.*"

*3.4. Providing Opportunities for Tribal Youth While Nurturing Their "True Self"*

"*But do I think there's a huge opportunity for tribal youth being involved with this and if Cal Poly is really serious about investing in the TEK education, I think you can bridge those together and collaborate. Maybe even get these kids you know, they're 15–16, the opportunity to work in the forest, and then by the time they're like graduate high school they're like, 'I want to go into like Resource Management.'*"

All participants (*n* = 13) viewed engaging tribal youth in the management of Goukdi'n positively through programs such as paid summer internships and job shadowing. There was a desire to teach them "*to respect the forests*" and also to "*keep them busy.*" Participants indicated that this opportunity could expose them to potential career paths which could lead them to local jobs. One participant shared that this opportunity could put in their mind something "*that they may have not thought about that as a career path . . . or something they could pursue in school.*" Another participant added that the university could support this by having, " . . . *some kind of stipend for them to be a part of it.*"

This opportunity could also be used to teach youth important life skills like how to gather necessities like firewood and locate water. Another participant shared that it was an opportunity to teach them about Wiyot Tribe cultural knowledge and teach them to "*...be stewards of the land again*" which another believed was an "*innate responsibility*" held by the youth. Another benefit of co-management was identified as learning the distinction between native and non-native plant species.

A participant shared the importance of having the opportunity to transfer traditional knowledge, "*I think it would be important, you know to get them back to their cultural stuff and, you know, all the things we used to do. There's a lot of knowledge being lost.*" However, this opportunity for knowledge transfer can also place responsibility on tribal youth and bring pressure, "*... But also, for themselves, we put a lot of pressure on them. Because they're young people, and because we know what can be lost, you know?*"

While all participants positively viewed this opportunity, they also shared that they hope that they could learn skills while nurturing their cultural values. There are unique pressures and expectations for tribal youth that should be considered. "*How do you take Indian kids, not pull the Indian out of them, try to make them something else. How do you . . . create a program that nurtures the true self?*". A younger participant shared that information from public schools conflicted with their cultural beliefs. They described this divergence of world views, "*I walked two worlds...because my family is very traditional.*"

Transportation was identified by 38% of participants (*n* = 5) as the biggest barrier to engaging tribal youth in the management of Goukdi'n. The remoteness of the reservation and the lack of resources was seen as two obstacles to overcome. When discussing this barrier, one participant shared, "*So I think the biggest one is like, ok, yeah, you have youth that wants to get involved but they don't have any transportation to get there.*"

## 4. Discussion

The use of semi-structured interviews served as a successful method of building bridges between the university and the Wiyot Tribe [49] and initiating dialogue [28] outside of formal government-to-government communication channels [24]. Participants appreciated that we sought the views of all Wiyot Tribal citizens and not exclusively members of the Wiyot Tribal Council. The specific views, opinions, and priorities of participants identified in this study provide valuable information and guidance to support university management if they continue to move toward developing a co-management partnership with the Wiyot Tribe. We utilized Decolonized Research Methodologies and focused on gaining insights from tribal members from different backgrounds and varying experiences

in natural resource management. While this study does not include a comparison of data collection methods, our use of semi-structured interviews served as an important step toward co-management and provided rich data to address our three research questions.

Our research questions were aimed at identifying the views on the co-management relationship, forest management objectives, and the importance of engaging with tribal youth in natural resource management. Although participants shared a variety of opinions on their expectations related to the research questions, four broad themes emerged from the data:

- Regaining access to ancestral forests can promote cultural knowledge and reconnect tribal members as stewards of the land;
- Conditions for achieving true co-management;
- Centering forest management activities on cultural resources, wildlife habitat, and the restoration of natural processes; and
- Providing opportunities for tribal youth while nurturing their "true self".

All 13 participants had positive views on the potential co-management partnership. Expectations of the university included clear communication, transparency, a willingness to listen, and keeping their word. The analysis of transcript data revealed that participants acknowledged that a forest may require active management to restore forest health, improve wildlife habitat, and mitigate wildfire risks. Management goals in Goukdi'n largely focused on promoting native plant and tree species that are of cultural importance to the tribe. Each participant positively viewed the engagement of tribal youth in natural resource management. However, they cautioned that native youth have unique pressures placed on them and need programs that both nurture their tribal cultural values and provide exposure to WEK skillsets. Here we discuss the main themes identified in the results section of this study. Then we will describe unexpected findings that emerged from the data.

### 4.1. Regaining Access to Ancestral Forests Can Promote Cultural Knowledge and Reconnect Tribal Members as Stewards of the Land

Our results align with findings from Diver [24] where the Karuk Tribe leveraged co-management as a way to increase access to cultural resources, build tribal capacity and increase the legitimacy of tribal management institutions. Recognition of rights to access and the preservation of cultural identity were also identified as key aspects in a study examining the co-management of freshwater resources by the Maori in New Zealand [1]. This study also supports the tribal co-management literature in identifying tribal culture [37] as an important component of a successful relationship.

### 4.2. Conditions for Achieving True Co-Management

Co-management literature has identified the integration of different ways of knowing as important to cogenerating knowledge [26,39,71,72]. Participants echoed this desire to use both TEK and WEK to develop management plans but, similar to a study in tribal co-management by Diver [24], they preferred that co-managers place greater importance on TEK over WEK. Similar to other studies, all participants indicated that they expect the university to respect tribal cultural values [1,17,18,24,31]. Incorporating and respecting TEK can help address the important cultural variable found to be important to co-management success [37].

The priorities of participants that we interviewed regarding co-management relationships supported the findings of Plummer et al. [45] in that collaboration and social learning were primary components of a successful partnership. Learning from each other was a common emergent theme in this study. Participants expect the university, as an educational institution, to be willing to learn from the tribe, just as they would learn from them. This process is described as social learning and is defined by iterative loops of assessment and re-adjusting governing and forest management approaches [21,68–70]. This iterative exchange of information has been seen as having the potential to create novel solutions to complex social-ecological issues [26,72]. Viewing the partnership as a process, participants'

expectations echoed previous research that highlights the need for partners to understand the long-term nature of co-management relationships [21,36,37].

While recognizing that there was a power imbalance due to the university holding title to the property as noted by Nadasdy [94], the participants in the study still expected that decision-making authority should be shared to reach what some researchers describe as complete tribal co-management [19,29,30]. Shared decision-making as an essential component of Adaptive Co-Management has been viewed by researchers as a rare occurrence in partnerships [31]. In one case study, the lack of decision-making authority was a reason that a tribal partner viewed co-management as a means to protect sacred lands, rather than an end goal of the partnership [32]. Participants in this study similarly viewed the relationship as a stepping stone toward other co-management relationships as well as building capacity for future forest management projects.

Additionally, studies have shown that expectations of clear communication and transparency on roles and responsibilities foster co-management [46,87] and are similarly supported by the results of this study.

### 4.3. Centering Forest Management Activities and Research on Wildlife Habitat and the Restoration of Natural Processes

The main interest of participants was a focus on forest management to improve wildlife habitat. This aligns with the primary purpose of the Goukdi'n forest conservation easement. Species-focused management has also been emphasized in a study investigating the integration of TEK and WEK [52]. In our interviews, this could be seen as one of the most important measurements of success when assessing forest management projects. Elsewhere, the integration of cultural values and views on the natural world has conflicted with agency management objectives around federal wildlife protection laws in the Klamath Basin [25].

The utilization of TEK as a primary component in research and management activities is supported by both federal [95] and state [15] executive actions and policies. At CPH, a recently released 5-year strategic plan [81] outlines goals of decolonizing research by valuing TEK and partnering with Indigenous communities. These "Visions" and "Core Values" signal that there are enabling environments [21,39] for tribal co-management on state-owned lands. Tribal partners' desire for TEK-generated research and management activities has also been observed in other tribal co-management studies [18,24].

Interview participants want management activities to focus on culturally important plant species, mitigating wildfire risk, and improving wildlife habitat by utilizing uneven-aged silviculture (Table 3). Uneven-aged silviculture does not remove all trees in harvest areas and leaves either single trees or groups of trees, with reductions in canopy cover to encourage a forest structure of varying age classes. Based on the results of this study, university managers should expect uneven-aged management to be the preferred silvicultural method of their tribal partner when conducting management activities in Goukdi'n.

**Table 3.** Examples of quotes provided by participants and how they could be translated into forest management objectives.

| Forest Management Objectives | Interview Quote |
| --- | --- |
| Manage for Uneven-aged Stand Structure | *"...you have to leave some so that the forest can create, again, its own, its own, its own ecosystems to care for all the different growing things underneath it. You know you have them big ones, to protect the medium ones, to protect the little ones. So, you can't cut them all."* |
| Conduct Extensive Botanical, Water quality, and Wildlife Surveys. | *"I'd love to see how many critters live on the land. I'd love to see what that water tastes like. Is it healthy water? Where does it come from? Is it good for drinking? I'd love to see what kind of lichens grow in the forest. I'd love to see these different...research opportunities."* |

**Table 3.** *Cont.*

| Forest Management Objectives | Interview Quote |
| --- | --- |
| Silviculture Promoting Native Understory Vegetation | *"You could plant with you know, a couple understory shrubs here and there as well, I know that sounds crazy . . . "* |
| Improve Wildlife Habitat | *" . . . and they did do some light logging. Not heavy, just light. And it was nice, it was really nice to see. So, that habitat plays a role to bring back wildlife."* |
| Develop (Interpretive) Trail System | *"It would be cool to see you know, a nice trail system that, that you know goes to different patches of interest."* |

*4.4. Providing Opportunities for Tribal Youth While Nurturing Their "True Self"*

Aligning with the Sowerwine et al. [17] study that sought the goals and objectives of a tribal research partner, we found that engaging tribal youth was an important component of co-management. Consistent with research investigating the education of tribal youth, our findings indicate that successful engagement includes the need for the university's partners to understand their tribal culture [53–57]. Therefore, co-management not only reconnects and provides access to forested lands but also provides an opportunity to train younger generations of tribal youth to empower the tribe's next generation of natural resource managers.

*4.5. Unexpected Findings*

While not deliberately investigated by asking questions, three unexpected results emerged from this study. First, returning the land to the Wiyot Tribe was only brought up by approximately 30% of the participants (*n* = 4). In none of these cases did the participant view the two actions, co-management, and land return, as mutually exclusive but would prefer the land to be returned to the tribe. The participants were, instead, concerned with restoring the forest, and improving wildlife habitat while focusing on Wiyot Tribe's cultural values and TEK. Second, many participants shared that they would like to see more Native American representation in the staff and curriculum being taught at CPH. When discussing this desire to have more representation at the university, it was also linked to creating a more inclusive environment for tribal students. Lastly, participants in this study not only sought to involve Wiyot Tribe membership in co-management but also mentioned including the non-native community. This finding is contradictory to a study of a co-management partnership between a tribe and the Bureau of Land Management (BLM) where the tribe sought to exclude and limit public access to a park for the protection of cultural resources [32]. This may be because the forest in this study is held privately rather than publicly and currently does not have many visitors.

*4.6. Limitations and Recommendations*

None of the participants had professional forestry experience which potentially limited the depth of their responses regarding forest management. While many of the participants had some experience in natural resource management in either report writing or an allied discipline (such as botany), they had limited understanding of common silvicultural pre-scriptions such as thinning and selection management, natural resource law, and forest operations. Without knowledge of different possible management options, participants were unable to describe, in detail, what types of activities they would like to see conducted in the forest.

Additionally, due to COVID-19-related restrictions on social gatherings and historic exclusion from their ancestral lands, only two of the participants in this study had recently visited this particular forest. This lack of knowledge of current forest conditions through-out Goukdi'n likely limited potential responses to questions about their goals for forest management. Further limitations existed in terms of participant sampling methodology and generalizability. For example, before conducting this research project, we had no contacts within the Wiyot Tribe and social capital was absent. This meant that we relied

on individuals with connections within the community to begin building relationships with Wiyot Tribe citizens. Conducting an interview project during a global pandemic in combination with limited social capital and with research fatigue within the community, we found difficulty reaching those in our potential pool of participants.

Finally, this study focused on the co-management priorities of only one of many Native American tribes and is therefore limited in its ability to be generalized. While this study provides detailed insight into tribal partners' views on co-management, due to the nonprobability sample these findings are not intended to be generalized to other tribes and tribal communities or other co-management cases [54]. Therefore, we recommend replicating our study, and also suggest that future studies of tribal co-management include interviews with both the institution and tribal partners to identify mutual goals and potential barriers to success.

## 5. Conclusions

Semi-structured interviews provided an opportunity to initiate meaningful dialogue between CPH and the Wiyot Tribe while answering our three research questions. This research aimed to identify: (i) views on co-management, (ii) forest management objectives, and (iii) the importance of engaging tribal youth in natural resource management. Based on a qualitative analysis of 13 interview transcripts we identified themes that informed these three research questions and found that (1) participants were optimistic about the potential partnership and anticipated that real benefits to the tribe could be born from the partnership, (2) forest management objectives and gauges of success were tied to the improvement of wildlife habitat and the promotion of culturally important native plant species, and (3) engaging tribal youth in the management of natural resources was unanimously viewed as beneficial. Our study suggests that the co-management of Goukdi'n can create benefits for the Wiyot Tribe by focusing on meeting their partnership requirements, managing expectations, and anticipating ways to address identified barriers. These findings are supported by the core components of ACM theory which could serve as a potential framework for the forming and building of a partnership between CPH and the Wiyot Tribe.

**Supplementary Materials:** The following supporting information can be downloaded at: https://www.mdpi.com/article/10.3390/f13122165/s1, Table S1: Interview Guide.

**Author Contributions:** Conceptualization, K.B.; Data curation, Z.J.E.; Formal analysis, Z.J.E., K.B. and M.J.D.; Funding acquisition, Z.J.E., K.B. and J.-P.B.; Methodology, Z.J.E., K.B. and M.J.D.; Project administration, J.-P.B.; Supervision, K.B. and J.-P.B.; Writing—original draft, Z.J.E.; Writing—review & editing, Z.J.E., K.B., M.J.D. and J.-P.B. All authors have read and agreed to the published version of the manuscript.

**Funding:** This research was funded by the Humboldt Area Foundation Donald Morris Hegy Memorial Fund and Joseph Sidney Woolford Fund, the Intertribal Timber Council's Native American Natural Resource Research Scholarship, Intertribal Student Services, and the Cal Poly Humboldt Sponsored Programs Foundation's Research and Creative Projects for Equity and Justice Grant #497 which included funding for the APC.

**Data Availability Statement:** Data presented in this study are available in this article. Raw data are protected and stored in a password protected folder according to IRB requirements.

**Acknowledgments:** We are indebted to the Wiyot Tribe interview participants, Natural Resources Department's (Shawir Darrudaluduk) Adam Canter and Hilanea Wilkinson, and the Tribal Council for approving and supporting this research. Lonyx Landry who works with the Cal Poly Humboldt Indian Natural Resources, Science, and Engineering Program and Diversity in STEM was essential in connecting the researchers to interview participants. We are grateful for the helpful peer-review comments provided by anonymous reviewers that improved this manuscript.

**Conflicts of Interest:** The authors declare no conflict of interest.

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
