# Peer review of "Listening to Indigenous Voices, Interests, and Priorities That Would Inform Tribal Co-Management of Natural Resources on a California State University Forest"

_forests, doi:10.3390/f13122165_

Round 1

Reviewer 1 Report

I found the paper well contextualized and structured. One of the good aspects of the paper is the explanation of the limitations.

 What kind of resources are extracted from the forests? For what purpose - subsistence or trade? Are there such practices? What is the cultural significance to the tribe? I would expect these questions to be answered. 

Author Response

Reviewer 1,

Thank you for your review of this journal article submission. I appreciate your positive comments on the structure and contextualization of the paper, as well as the limitations section.

I also thank you for your questions regarding the article and will address them here and indicate how I have answered them through minor revisions of the document.

Currently there are no resource being extracted from the forest and is a new acquisition for the university. This study sought to understand what types of resources the Wiyot Tribe would like to extract from the forest. From the interviews, we have reason to believe that these resources would likely be used for cultural purposes, including subsistence.

While the tribe does practice gathering of materials and foods from the forest, the Goukdi’n (Jacoby Creek) property does not currently have such practices occurring, to the knowledge of the researchers.

The cultural significance of this property is large because of the proximity to Wigi, or Humboldt Bay. Jacoby Creek, the creek that runs through the property, supports salmon and other aquatic species at its lower reaches (below the property boundary). This property is located at the headwaters of that stream and therefore has a large impact on the lower habitats for culturally important species. We have added a short passage that shares some of the culturally important resources used by the Wiyot Tribe in the forest (Lines 160-167).

Thank you again for your review of this journal article submission.

All the best,

Zachary J. Erickson

Reviewer 2 Report

The paper is interesting, however, my opinion is that it could be improved with more surveys. Also, it could be statement how the this kind of surveys will be taken in account in the forest management.

Author Response

Reviewer 2,

Thank you for your review and questions regarding this journal article submission.

Here I will address your two questions/suggestions you brought up during the review.

We agree that more surveys may have provided some improvement on the study. We found severe limitations in accessing the tribal community due to COVID-19 related lockdowns and restrictions. The Wiyot Tribe is also a relatively small sized Native American Tribe. Although an increase in the sample size was desired, we feel confident that we reached our goal of “theoretical saturation”. The responses in interviews became repetitive and no new information regarding our research questions was arising from subsequent interviews. We refer to this method of assessing our studies sample size on Line 274. We also sought transparency with the limitations of the sample size beginning on Line 770.

In regards to how this survey will be taken into account in the forest management, the project itself serves as “…an opportunity to initiate meaningful dialogue between CPH and the Wiyot Tribe” (Line 796). The forest is newly acquired and the results of this study identified potential management objectives including the promotion of “…wildlife habitat and the promotion of culturally important native plant species…” (Line 798).

We thank you again for your review and accompanied questions and suggestions.

All the best,

Zachary J. Erickson